# On a Class of Tensor Markov Fields

**DOI:** 10.3390/e22040451

**Published:** 2020-04-16

**Authors:** Enrique Hernández-Lemus

**Affiliations:** 1Computational Genomics Division, National Institute of Genomic Medicine, 14610 Mexico City, Mexico; ehernandez@inmegen.gob.mx; Tel.: +52-55-5350-1970; 2Centro de Ciencias de la Complejidad, Universidad Nacional Autónoma de México, 04510 Mexico City, Mexico

**Keywords:** Markov random fields, probabilistic graphical models, multilayer networks

## Abstract

Here, we introduce a class of Tensor Markov Fields intended as probabilistic graphical models from random variables spanned over multiplexed contexts. These fields are an extension of Markov Random Fields for tensor-valued random variables. By extending the results of Dobruschin, Hammersley and Clifford to such tensor valued fields, we proved that tensor Markov fields are indeed Gibbs fields, whenever strictly positive probability measures are considered. Hence, there is a direct relationship with many results from theoretical statistical mechanics. We showed how this class of Markov fields it can be built based on a statistical dependency structures inferred on information theoretical grounds over empirical data. Thus, aside from purely theoretical interest, the Tensor Markov Fields described here may be useful for mathematical modeling and data analysis due to their intrinsic simplicity and generality.

## 1. General Definitions

Here, we introduce Tensor Markov Fields, i.e., Markov random fields [1,2] over tensor spaces. Tensor Markov Fields (TMFs) represent the joint probability distribution for a set of tensor-valued random variables.

Let $X=X_{\alpha }^{\beta }$ be one of such tensor-valued random variables. Here $X_{i}^{j}\in X$ may represent either a variable i∈α, that may exist in a given context or layer j∈β (giving rise to a class of so-called multilayer graphical models or multilayer networks) or a single tensor-valued quantity $X_{i}^{j}$. A TMF will be an undirected multilayer graph representing the statistical dependency structure of *X* as given by the joint probability distribution P(X).

As an extension of the case of Markov random fields, a TMF is a multilayer graph $\hat{G}=\left(V,E\right)$ formed by a set *V* of vertices or nodes (the $X_{i}^{j}$’s) and a set E⊆V×V of edges connecting the nodes, either on the same *layer* or through different layers (Figure 1). The set of edges represents a neighborhood law *N* stating which vertex is connected (dependent) to which other vertex in the multilayer graph. With this in mind, a TMF can be also represented (slightly abusing notation) as $\hat{G}=\left(V,N\right)$. The set of neighbors of a given point $X_{i}^{j}$ will be denoted $N_{X_{i}^{j}}$.

### 1.1. Configuration

It is possible to assign to each point in the multilayer graph, one of a finite set *S* of labels. Such assignment will be called a *configuration*. We will assign probability measures to the set Ω of all possible configurations ω. Hence, $\omega_{A}$ represents the configuration ω restricted to the subset *A* of *V*. It is possible to think of $\omega_{A}$ as a configuration on the smaller multilayer graph ${\hat{G}}_{A}$ restricting *V* to points of *A* (Figure 2).

### 1.2. Local Characteristics

It is also possible to extend the notion of *local characteristics* from MRFs. The local characteristics of a probability measure P defined on Ω are the conditional probabilities of the form:(1)$$P\left(\omega_{t}\;|\;\omega_{T\backslash t}\right)=P\left(\omega_{t}\;|\;\omega_{N_{t}}\right)$$
i.e., the probability that the point *t* is assigned the value $\omega_{t}$, given the values at all other points of the multilayer graph. In order to make explicit the tensorial nature of the multilayer graph $\hat{G}$, let us re-write Equation (Equation 1). Let us also recall the fact that the probability measure will define a tensor Markov random field (a TMF) if the local characteristics depend only of the knowledge of the outcomes at neighboring points, i.e., if for every ω
(2)$$P\left(\omega_{X_{i}^{j}}\;|\;\omega_{\hat{G}\backslash X_{i}^{j}}\right)=P\left(\omega_{X_{i}^{j}}\;|\;\omega_{N_{X_{i}^{j}}}\right)$$

### 1.3. Cliques

Given an arbitrary graph (or in the present case a multilayer graph), we shall say that a set of points *C* is a *clique* if every pair of points in *C* are neighbors (see Figure 3). This definition includes the empty set as a clique. A clique is thus a set whose *induced subgraph* is complete, for this reason cliques are also called *complete induced subgraphs* or *maximal subgraphs* (although these latter term may be ambiguous).

### 1.4. Configuration Potentials

A *potential*
η is a way to assign a number $\eta_{A}\left(\omega \right)$ to every subconfiguration $\omega_{A}$ of a configuration ω in the multilayer graph $\hat{G}$. Given a potential, we shall say that it defines (or better, induces) a *dimensionless energy*
U(ω) on the set of all configurations ω by
(3)$$U\left(\omega \right)=\sum_{A}\eta_{A}\left(\omega \right)$$

In the preceeding expression, for fixed ω, the sum is taken over all subsets A⊆V including the empty set. We can define a probability measure, called the *Gibbs measure induced* by *U* as
(4)$$P\left(\omega \right)=\frac{e^{-U(\omega )}}{Z}$$
with *Z* a normalization constant called the *partition function*.
(5)$$Z=\sum_{\omega }e^{-U(\omega )}$$

In physics, the term *potential* is often used in connection with the so-called potential energies. Physicists often call $\eta_{A}$ a *dimensionless potential energy*, and they call $\phi_{A}=e^{-\eta_{A}}$ a potential.

Equations (Equation 4) and (Equation 5) can be thus rewritten as:(6)$$P\left(\omega \right)=\frac{\prod_{A}\phi_{A}\left(\omega \right)}{Z}$$
(7)$$Z=\sum_{\omega }\prod_{A}\phi_{A}\left(\omega \right)$$

Since this latter use is more common in probability and graph theory, we will refer to Equations (Equation 6) and (Equation 7) as the definitions of Gibbs measure and partition function (respectively) unless otherwise stated.

### 1.5. Gibbs Fields

A potential is called a nearest neighbor Gibbs potential if $\phi_{A}\left(\omega \right)=1$ whenever *A* is not a clique. It is customary to refer as a *Gibbs measure* to a measure induced by a nearest neighbor Gibbs potential. However, it is possible to define more general Gibbs measures by considering other types of potentials.

The inclusion of all cliques in the calculation of the Gibbs measure is necessary to establish the equivalence between Gibbs random fields and Markov random fields. Let us see how a nearest neighbor Gibbs measure on a multilayer graph determines a TMF.

Let P(ω) be a probability measure determined on Ω by a nearest neighbor Gibbs potential ϕ:(8)$$P\left(\omega \right)=\frac{\prod_{C}\phi_{C}\left(\omega \right)}{Z}$$

With the product taken over all cliques *C* on the multilayer graph $\hat{G}$. Then,
(9)$$P\left(\omega_{X_{i}^{j}}|\omega_{\hat{G}\backslash X_{i}^{j}}\right)=\frac{P(\omega )}{\sum_{\omega^{\prime }}P\left(\omega^{\prime }\right)}$$

Here $\omega^{\prime }$ is any configuration which agrees with ω at all points except $X_{i}^{j}$.
(10)$$P\left(\omega_{X_{i}^{j}}|\omega_{\hat{G}\backslash X_{i}^{j}}\right)=\frac{\prod_{C}\phi_{C}\left(\omega \right)}{\sum_{\omega^{\prime }}\prod_{C}\phi_{C}\left(\omega^{\prime }\right)}$$

For any clique *C* that does not contain $X_{i}^{j}$, $\phi_{C}\left(\omega \right)=\phi_{C}\left(\omega^{\prime }\right)$, So that all the terms that correspond to cliques that do not contain the point $X_{i}^{j}$ cancel both from the numerator and the denominator in Equation (Equation 10), therefore this probability depends only on the values $x_{i}^{j}$ at $X_{i}^{j}$ and its neighbors. P defines thus a TMF.

A more general proof of this equivalence is given by Hammersley-Clifford theorem that will be presented in the following section.

## 2. Extended Hammersley Clifford Theorem

Here we will outline a proof for an extension of Hammersley-Clifford theorem for Tensor Markov Fields (i.e., we will show that a Tensor Markov Field is equivalent to a Tensor Gibbs Field).

Let $\hat{G}=\left(V,N\right)$ be a multilayer graph representing a TMF as defined in the previous section. With $V=X_{\alpha }^{\beta }=\left\{X_{i}^{j}\right\}$, a set of vertices over a tensor field and *N* a neighborhood law that connects vertices over this tensor field. The field $\hat{G}$ obeys the following neighborhood law given its Markov property (see Equation (Equation 2))
(11)$$P\left(X_{i}^{j}|X_{\hat{G}\backslash X_{i}^{j}}\right)=P\left(X_{i}^{j}|X_{N_{i}^{j}}\right)$$

Here $X_{N_{i}^{j}}$ is any neighbor of $X_{i}^{j}$. The Hammersley-Clifford theorem states that a MRF is also a local Gibbs field. In the case of a TMF we have the following expression:(12)$$P\left(X\right)=\frac{1}{Z}\prod_{c\in C_{\hat{G}}}\phi_{c}\left(X_{c}\right)$$

In order to prove the equivalence of Equations (Equation 11) and (Equation 12), we will first bult a deductive (backward direction) part of the proof to be complemented with a constructive (forward direction) part as presented in the following subsections.

### 2.1. Backward Direction

Let us consider Equation (Equation 11) at the light of Bayes’ theorem:(13)$$P\left(X_{i}^{j}|X_{\hat{G}\backslash X_{i}^{j}}\right)=\frac{P(X_{i}^{j},X_{N_{i}^{j}})}{P(X_{N_{i}^{j}})}$$

Using a clique-approach to calculate the joint and marginal probabilities (see next subsection to support the following statement):(14)$$P\left(X_{i}^{j}|X_{\hat{G}\backslash X_{i}^{j}}\right)=\frac{\sum_{\hat{G}\backslash D_{i}^{j}}\prod_{c\in C_{\hat{G}}}\phi_{c}\left(X_{c}\right)}{\sum_{X_{i}^{j}}\sum_{\hat{G}\backslash D_{i}^{j}}\prod_{c\in C_{\hat{G}}}\phi_{c}\left(X_{c}\right)}$$

Let us split the product $\prod_{c\in C_{\hat{G}}}\phi_{c}\left(X_{c}\right)$ into two products, one over the set of cliques that contain $X_{i}^{j}$ (let us call it $C_{i}^{j}$) and another set formed by cliques not containing $X_{i}^{j}$ (let us call it $R_{i}^{j}$):(15)$$P\left(X_{i}^{j}|X_{\hat{G}\backslash X_{i}^{j}}\right)=\frac{\sum_{\hat{G}\backslash D_{i}^{j}}\prod_{c\in C_{i}^{j}}\phi_{c}\left(X_{c}\right)\;\prod_{c\in R_{i}^{j}}\phi_{c}\left(X_{c}\right)}{\sum_{X_{i}^{j}}\sum_{\hat{G}\backslash D_{i}^{j}}\prod_{c\in C_{i}^{j}}\phi_{c}\left(X_{c}\right)\;\prod_{c\in R_{i}^{j}}\phi_{c}\left(X_{c}\right)}$$

Factoring out the terms depending on $X_{i}^{j}$ (that do not contribute to cliques in the domain $\hat{G}\backslash X_{i}^{j}$):(16)$$P\left(X_{i}^{j}|X_{\hat{G}\backslash X_{i}^{j}}\right)=\frac{\prod_{c\in C_{i}^{j}}\phi_{c}\left(X_{c}\right)\;\sum_{\hat{G}\backslash D_{i}^{j}}\prod_{c\in R_{i}^{j}}\phi_{c}\left(X_{c}\right)}{\sum_{X_{i}^{j}}\prod_{c\in C_{i}^{j}}\phi_{c}\left(X_{c}\right)\;\sum_{\hat{G}\backslash D_{i}^{j}}\prod_{c\in R_{i}^{j}}\phi_{c}\left(X_{c}\right)}$$

The term $\sum_{\hat{G}\backslash D_{i}^{j}}\prod_{c\in R_{i}^{j}}\phi_{c}\left(X_{c}\right)$ does not involve $X_{i}^{j}$ (by construction) so, it can be factored out from the summation over $X_{i}^{j}$ in the denominator.
(17)$$P\left(X_{i}^{j}|X_{\hat{G}\backslash X_{i}^{j}}\right)=\frac{\prod_{c\in C_{i}^{j}}\phi_{c}\left(X_{c}\right)\sum_{\hat{G}\backslash D_{i}^{j}}\prod_{c\in R_{i}^{j}}\phi_{c}\left(X_{c}\right)}{\sum_{\hat{G}\backslash D_{i}^{j}}\prod_{c\in R_{i}^{j}}\phi_{c}\left(X_{c}\right)\;\sum_{X_{i}^{j}}\prod_{c\in C_{i}^{j}}\phi_{c}\left(X_{c}\right)}$$

We can cancel the term in the numerator and denominator:(18)$$P\left(X_{i}^{j}|X_{\hat{G}\backslash X_{i}^{j}}\right)=\frac{\prod_{c\in C_{i}^{j}}\phi_{c}\left(X_{c}\right)}{\sum_{X_{i}^{j}}\prod_{c\in C_{i}^{j}}\phi_{c}\left(X_{c}\right)}$$

Then we multiply by $\frac{\prod_{c\in R_{i}^{j}}\phi_{c}\left(X_{c}\right)}{\prod_{c\in R_{i}^{j}}\phi_{c}\left(X_{c}\right)}$
(19)$$P\left(X_{i}^{j}|X_{\hat{G}\backslash X_{i}^{j}}\right)=\frac{\prod_{c\in C_{i}^{j}}\phi_{c}\left(X_{c}\right)\prod_{c\in R_{i}^{j}}\phi_{c}\left(X_{c}\right)}{\sum_{X_{i}^{j}}\prod_{c\in C_{i}^{j}}\phi_{c}\left(X_{c}\right)\prod_{c\in R_{i}^{j}}\phi_{c}\left(X_{c}\right)}$$

Remembering that $C_{i}^{j}⋃R_{i}^{j}=C_{\hat{G}}$,
(20)$$P\left(X_{i}^{j}|X_{\hat{G}\backslash X_{i}^{j}}\right)=\frac{\prod_{c\in \hat{G}}\phi_{c}\left(X_{c}\right)}{\sum_{X_{i}^{j}}\prod_{c\in \hat{G}}\phi_{c}\left(X_{c}\right)}$$

Equation (Equation 20) is nothing but the definition of a local Gibbs Tensor Field (Equation (Equation 12)).

### 2.2. Forward Direction

In this subsection we will show how to express the clique potential functions $\phi_{c}\left(X_{c}\right)$, given the joint probability distribution over the tensor field and the Markov property.

Consider any subset $\sigma \subset \hat{G}$ of the multilayer graph $\hat{G}$. We define a candidate potential function (following Möbius inversion lemma) [3] as follows:(21)$$f_{\sigma }\left(X_{\sigma }=x_{\sigma }\right)=\prod_{\zeta \subset \sigma }\;P{\left(X_{\zeta }=x_{\zeta },X_{\hat{G}\backslash \zeta }=0\right)}^{-1^{\left|\sigma |-|\zeta \right|}}$$

In order for $f_{\sigma }$ to be a proper clique potential, it must satisfy the following two conditions:(i)$\prod_{\sigma \subset \hat{G}}\;f_{\sigma }\left(X_{\sigma }\right)=P\left(X\right)$(ii)$f_{\sigma }\left(X_{\sigma }\right)=1$ whenever σ is not a clique

To prove (i), we need to show that all factors in $f_{\sigma }\left(X_{\sigma }=x_{\sigma }\right)$ cancel out, except for P(X).

To do this, it will be useful to consider the following *combinatorial expansion of zero*:(22)$$0={\left(1-1\right)}^{K}=C_{0}^{K}-C_{1}^{K}+C_{2}^{K}+\cdots +{\left(-1\right)}^{K}C_{K}^{K}$$

Here, of course $C_{B}^{A}$ is the number of combinations of B elements from an A-element set.

Let us consider any subset ζ of $\hat{G}$. Let us consider a factor $\Delta =P(X_{\zeta }=x_{\zeta },X_{\hat{G}\backslash \zeta }=0)$. For the case of $f\zeta (X_{\zeta })$ it occurs as $\Delta^{-1^{0}}=\Delta$. Such factor also occurs in subsets containg ζ and other additional elements. If it includes ζ and one additional element, there are $C_{1}^{|\hat{G}\left|-|\zeta \right|}$ such functions. The additional element creates an inverse factor $\Delta^{-1^{1}}=\Delta^{-1}$. The functions over subsets containg ζ and two additional elements contributes with a factor $\Delta^{-1^{2}}=\Delta^{1}=\Delta$. If we continue this process and consider Equation (Equation 22), it is evident that all odd cardinality difference terms cancel out with all even cardinality difference terms so that the only remaining factor corresponds to $\zeta =\hat{G}$ equal to P(X) thus fulfilling condition (i).

In order to show how condition (ii) is fulfilled, we will need to use the Markov property of TMFs. Let us consider $\sigma *\subset \hat{G}$ that is not a clique. Then it will be possible to find two nodes $X_{i}^{h}$ and $X_{j}^{k}$ in σ* that are not connected to each other. Let us recall Equation (Equation 21):(23)$$f_{\sigma }\left(X_{\sigma }*=x_{\sigma }*\right)=\prod_{\zeta \subset \sigma *}\;P{\left(X_{\zeta }=x_{\zeta },X_{\hat{G}\backslash \zeta }=0\right)}^{-1^{\left|\sigma *|-|\zeta \right|}}$$

An arbitrary subset ζ may belong to any of the following classes: (i) ζ=ω a generic subset of σ; (ii) $\zeta =\omega \cup \{X_{i}^{h}\}$; (iii) $\zeta =\omega \cup \{X_{j}^{k}\}$ or (iv) $\zeta =\omega \cup \{X_{i}^{h},X_{j}^{k}\}$. If we write down Equation (Equation 23) factored down to these contributions we get:(24)$$f_{\sigma }\left(X_{\sigma }*=x_{\sigma }*\right)=\prod_{\omega \subset \sigma *\backslash \{X_{i}^{h},X_{j}^{k}\}}{\left[\frac{P\left(X_{\omega },X_{\hat{G}\backslash \omega }=0\right)P\left(X_{\omega \cup \{X_{i}^{h},X_{j}^{k}\}},X_{\hat{G}\backslash \omega \cup \left\{X_{i}^{h},X_{j}^{k}\right\}}=0\right)}{P\left(X_{\omega \cup \{X_{i}^{h}\}},X_{\hat{G}\backslash \omega \cup \left\{X_{i}^{h}\right\}}=0\right)P\left(X_{\omega \cup \{X_{j}^{k}\}},X_{\hat{G}\backslash \omega \cup \left\{X_{j}^{k}\right\}}=0\right)}\right]}^{-1^{\left|\sigma *|-|\zeta \right|}}$$

Let us consider two of the factors in Equation (Equation 24) at the light of Bayes’ theorem:(25)$$\frac{P(X_{\omega },X_{\hat{G}\backslash \omega }=0)}{P(X_{\omega \cup \{X_{i}^{h}\}},X_{\hat{G}\backslash \omega \cup \left\{X_{i}^{h}\right\}}=0)}=\frac{P\left(X_{\left\{X_{i}^{h}\right\}}=0|X_{\left\{X_{j}^{k}\right\}}=0,X_{\omega },X_{\hat{G}\backslash \omega \cup \left\{X_{i}^{h},X_{j}^{k}\right\}}=0\right)P\left(X_{\left\{X_{j}^{k}\right\}}=0,X_{\omega },X_{\hat{G}\backslash \omega \cup \left\{X_{i}^{h},X_{j}^{k}\right\}}=0\right)}{P\left(X_{\left\{X_{i}^{h}\right\}}|X_{\left\{X_{j}^{k}\right\}}=0,X_{\omega },X_{\hat{G}\backslash \omega \cup \left\{X_{i}^{h},X_{j}^{k}\right\}}=0\right)P\left(X_{\left\{X_{j}^{k}\right\}}=0,X_{\omega },X_{\hat{G}\backslash \omega \cup \left\{X_{i}^{h},X_{j}^{k}\right\}}=0\right)}$$

We can notice that the priors in the numerator and denominator of Equation (Equation 25) are the same. We can then cancell them out. Since by definition $X_{i}^{h}$ and $X_{j}^{k}$ are conditionally independent given the rest of the multilayer graph, we can also replace the default value $X_{j}^{k}=0$ for $X_{j}^{k}$ instead.
(26)$$\frac{P(X_{\omega },X_{\hat{G}\backslash \omega }=0)}{P(X_{\omega \cup \{X_{i}^{h}\}},X_{\hat{G}\backslash \omega \cup \left\{X_{i}^{h}\right\}}=0)}=\frac{P\left(X_{\left\{X_{i}^{h}\right\}}=0|X_{\left\{X_{j}^{k}\right\}},X_{\omega },X_{\hat{G}\backslash \omega \cup \left\{X_{i}^{h},X_{j}^{k}\right\}}=0\right)\;P\left(X_{\left\{X_{j}^{k}\right\}},X_{\omega },X_{\hat{G}\backslash \omega \cup \left\{X_{i}^{h},X_{j}^{k}\right\}}=0\right)}{P\left(X_{\left\{X_{i}^{h}\right\}}|X_{\left\{X_{j}^{k}\right\}},X_{\omega },X_{\hat{G}\backslash \omega \cup \left\{X_{i}^{h},X_{j}^{k}\right\}}=0\right)\;P\left(X_{\left\{X_{j}^{k}\right\}},X_{\omega },X_{\hat{G}\backslash \omega \cup \left\{X_{i}^{h},X_{j}^{k}\right\}}=0\right)}$$

Since $X_{i}^{h}$ and $X_{j}^{k}$ are conditionally independent given the rest of the multilayer graph, we can also replace the condition for $X_{j}^{k}$ with any other, without affecting $X_{i}^{h}$. By adjusting this prior *conveniently*, we can write out:(27)$$\frac{P(X_{\omega },X_{\hat{G}\backslash \omega }=0)}{P(X_{\omega \cup \{X_{i}^{h}\}},X_{\hat{G}\backslash \omega \cup \left\{X_{i}^{h}\right\}}=0)}=\frac{P(X_{\omega \cup \{X_{j}^{k}\}},X_{\hat{G}\backslash \omega \cup \left\{X_{j}^{k}\right\}}=0)}{P(X_{\omega \cup \{X_{i}^{h},X_{j}^{k}\}},X_{\hat{G}\backslash \omega \cup \left\{X_{i}^{h},X_{j}^{k}\right\}}=0)}$$

By substituting Equation (Equation 27) into Equation (Equation 24) we get (condition (ii)):(28)$$f_{\sigma }*\left(X_{\sigma }*\right)=1$$

## 3. An Information-Theoretical Class of Tensor Markov Fields

Let us consider again the set of tensor-valued random variables $X=X_{\alpha }^{\beta }$. It is possible to calculate, for every duplex in *X*, the mutual information function I(·,·) [4]:(29)$$I\left(X_{i}^{h},X_{j}^{k}\right)=\sum_{\Omega }\sum_{\Omega^{\prime }}\;p\left(X_{i}^{h},X_{j}^{k}\right)\;\log\frac{p(X_{i}^{h},X_{j}^{k})}{p\left(X_{i}^{h}\right)\;p\left(X_{j}^{k}\right)}$$

Let us consider a multilayer graph scenario. From now on, the indices i,j will refer to the random variables, whereas h,k will be indices for the layers. Ω and $\Omega^{\prime }$ are the respective sampling spaces (that may, of course, be equal). In order to discard self-information, let us define the *off-diagonal mutual information* as follows:(30)$$I^{†}\left(X_{i}^{h},X_{j}^{k}\right)=I\left(X_{i}^{h},X_{j}^{k}\right)\times \left(1-\delta_{X_{i}^{h}X_{j}^{k}}\right)$$

With the bi-delta function $\delta_{X_{i}^{h}X_{j}^{k}}$ defined as:(31)$$\delta_{X_{i}^{h}X_{j}^{k}}=\left\{\begin{array}{cc} 1, & \mathrm{if}\;i=j\;\mathrm{and}\;h=k\\ 0, & \mathrm{otherwise} \end{array}\right.$$

By having the complete set of off-diagonal mutual information functions for all the random variables and layers, it is possible to define the following hyper-matrix elements:(32)$$A_{ij}^{hk}=\Theta \left[I^{†}\left(X_{i}^{h},X_{j}^{k}\right)\;-\;I_{0}\right]$$
as well as:(33)$$W_{ij}^{hk}=A_{ij}^{hk}\circ I^{†}\left(X_{i}^{h},X_{j}^{k}\right)$$

Here Θ[·] is Heavyside’s function and $I_{0}$ is a lower bound for mutual information (a threshold) to be considered *significant*.

We call $A_{ij}^{hk}$ and $W_{ij}^{hk}$ the *adjacency hypermatrix* and the *strength hypermatrix* respectively (notice that ∘ in Equation (Equation 33) represents the product of a scalar times a hypermatrix). The adjacency hyper-matrix and the strength hyper-matrix define the (unweighted and weighted, respectively) neighborhood law of the associated TMF, hence the statistical dependency structure for the set of random variables and contexts (layers).

Although the adjacency and strength hypermatrices are indeed, proper representations of the undirected (unweighted and weighted) dependency structure of P(X), it has been considered advantageous to embed them into a tensor linear structure, in order to be able to work out some of the mathematical properties of such fields relying on the methods of tensor algebra. One relevant proposal in this regard, has been advanced by De Domenico and collaborators, in the context of multilayer networks.

Following the ideas of De Domenico and co-workers [5], we introduce the unweighted and weighted adjacency 4-tensors (respectively) as follows:(34)$$A=\sum_{h,k=1}^{L}\sum_{i,j=1}^{N}A_{ij}^{hk}⊗\xi_{\beta \delta }^{\alpha \gamma }$$
(35)$$W=\sum_{h,k=1}^{L}\sum_{i,j=1}^{N}W_{ij}^{hk}⊗\xi_{\beta \delta }^{\alpha \gamma }$$

Here, $\xi_{\beta \delta }^{\alpha \gamma }=\xi_{\beta \delta }^{\alpha \gamma }\left[ijhk\right]$ is a unit four-tensor whose role is to provide the hypermatrices with the desired linear properties (projections, contractions, etc.). Square brackets indicate that the indices i,j,h and *k* belong to the α,β,γ and δ dimensions and ⊗ represents a form of a *tensor matricization* product (i.e., the one producing a 4-tensor out of a 4-index hypermatrix times a unitary 4-tensor).

### 3.1. Conditional Independence in Tensor Markov Fields

In order to discuss the conditional independence structure induced by the present class of TMFs, let us analyze Equation (Equation 32). As already mentioned, the hyper-adjacency matrix $A_{ij}^{hk}$ represents the neigborhood law (as given by the Markov property) on the multilayer graph $\hat{G}$ (i.e., the TMF). Every non-zero entry on this hypermatrix represents a statistical dependence relation between two elements on *X*. The conditional dependence structure on TMFs inferred from mutual information measures via Equation (Equation 32) are related not only to the statistical independence conditions (as given by a zero mutual information measure between two elements), but also to the lower bound $I_{0}$ and in general to the dependency structure of the whole multilayer graph.

The definition of conditional independence (CI) for tensor random variables is as follows:(36)$$\left(X_{i}^{h}⊥\;\;\;⊥X_{j}^{k}\right)|X_{l}^{m}\Leftrightarrow F_{X_{i}^{h},X_{j}^{k}|X_{l}^{m}=X_{l}^{m}*}\left(X_{i}^{h}*,X_{j}^{k}*\right)=F_{X_{i}^{h}|X_{l}^{m}=X_{l}^{m}*}\left(X_{i}^{h}*\right)\cdot F_{X_{j}^{k}|X_{l}^{m}=X_{l}^{m}*}\left(X_{j}^{k}*\right)$$
$\forall \;\;X_{i}^{h},X_{j}^{k},X_{l}^{m}\in X$.

Here ⊥⊥ represents conditional independence between two random variables, were $F_{X_{i}^{h},X_{j}^{k}|X_{l}^{m}=X_{l}^{m}*}\left(X_{i}^{h}*,X_{j}^{k}*\right)=Pr\left(X_{i}^{h}\leq X_{i}^{h}*,X_{j}^{k}\leq X_{j}^{k}*|X_{l}^{m}=X_{l}^{m}*\right)$ is the joint conditional cumulative distribution of $X_{i}^{h}$ and $X_{j}^{k}$ given $X_{l}^{m}$ and $X_{i}^{h}*$, $X_{j}^{k}*$ and $X_{l}^{m}*$ are realization events of the corresponding random variables.

In the case of MRFs (and by extension TMFs), CI is defined by means of (multi)graph separation: in this sense we say that $X_{i}^{h}⊥\;\;\;{⊥}_{\hat{G}}X_{j}^{k}|X_{l}^{m}$ iff $X_{l}^{m}$ separates $X_{i}^{h}$ from $X_{j}^{k}$ in the multilayer graph $\hat{G}$. This means that if we remove node $X_{l}^{m}$ there are no undirected paths from $X_{i}^{h}$ to $X_{j}^{k}$ in $\hat{G}$.

Conditional independence in random fields is often considered in terms of subsets of *V*. Let *A*, *B* and *C* be three subsets of *V*. The statement $X_{A}⊥\;\;\;{⊥}_{\hat{G}}X_{B}|X_{C}$, which holds only iff *C* separates *A* from *B* in the multilayer graph $\hat{G}$, meaning that if we remove all vertices in *C* there will be no paths connecting any vertex in *A* to any vertex in *B* is called the *global Markov property* of TMFs.

The smallest set of vertices that renders a vertex $X_{i}^{h}$ conditionally independent of all other vertices in the multilayer graph is called its *Markov blanket*, denoted $mb(X_{i}^{h})$. If we define the *closure* of a node $X_{i}^{h}$ as $C(X_{i}^{h})$ then $X_{i}^{h}⊥\;\;\;⊥\hat{G}\backslash C\left(X_{i}^{h}\right)|mb\left(X_{i}^{h}\right)$.

It is possible to show that in a TMF, the Markov blanket of a vertex is its set of first neighbors. This is called the *undirected local Markov property*. Starting from the local Markov property it is possible to show that two vertices $X_{i}^{h}$ and $X_{j}^{k}$ are conditionally independent given the rest if there is no direct edge between them. This has been called the *pairwise Markov property*.

If we denote by ${\hat{G}}_{X_{i}^{h}\to X_{j}^{k}}$ the set of undirected paths in the multilayer graph $\hat{G}$ connecting vertices $X_{i}^{h}$ and $X_{j}^{k}$, then the pairwise Markov property of a TMF can be stated as:(37)$$X_{i}^{h}⊥\;\;\;⊥X_{j}^{k}|\hat{G}\backslash \left\{X_{i}^{h},X_{j}^{k}\right\}\;\Leftrightarrow \;{\hat{G}}_{X_{i}^{h}\to X_{j}^{k}}=\emptyset$$

It is clear that the global Markov property implies the local Markov property which in turn implies the pairwise Markov property. For systems with positive definity probability densities, it has been probed (in the case of MRFs) that pairwise Markov actually implied global Markov (See [6] p. 119 for a proof). For the present extension this is important since it is easier to assess pairwise conditional independence statements.

### 3.2. Indepence Maps

Let $I_{\hat{G}}$ denote the set of all conditional independence relations encoded by the multilayer graph $\hat{G}$ (i.e., those CI relations given by the Global Markov property). Let $I_{P}$ be the set of all CI relations implied by the probability distribution $P(X_{i}^{j})$. A multilayer graph $\hat{G}$ will be called an *independence map* (*I-map*) for a probability distribution $P(X_{i}^{j})$, if all CI relations implied by $\hat{G}$ hold for $P(X_{i}^{j})$, i.e., $I_{\hat{G}}\subseteq I_{P}$ [6].

The converse statement is not necessarily true, i.e., there may be some CI relations implied by $P(X_{i}^{j})$ that are not encoded in the multilayer graph $\hat{G}$. We may be usually interested in *minimal I-maps*, i.e., I-maps from which none of the edges could be removed without destroying its CI properties.

Every distribution has a unique minimal I-map (and a given graph representation). Let $P(X_{i}^{j})>0$. Let ${\hat{G}}^{†}$ be the multilayer graph obtained by introducing edges between all pairs of vertices $X_{i}^{h}$, $X_{j}^{k}$ such that $X_{i}^{h}⊥\;\;\;⊥X_{j}^{k}|X\backslash \left\{X_{i}^{h},X_{j}^{k}\right\}$, then ${\hat{G}}^{†}$ is the unique minimal I-map. We call $\hat{G}$ a *perfect map* of P when there is no dependencies $\hat{G}$ which are not indicated by P, i.e., $I_{\hat{G}}=I_{P}$ [6].

### 3.3. Conditional Independence Tests

Conditional independence tests are useful to evaluate whether CI conditions apply either exactly or in the case of applications under a certain bounded error. In order to be able to write down expressions for C.I. tests let us introduce the following *conditional kernels* [7]:(38)$$C_{A}\left(B\right)=P\left(B|A\right)=\frac{P(AB)}{P(A)}$$
as well as their generalized recursive relations:(39)$$C_{ABC}\left(D\right)=C_{AB}\left(D|C\right)=\frac{C_{AB}\left(CD\right)}{C_{AB}\left(C\right)}$$

The conditional probability of $X_{h}^{k}$ given $X_{i}^{j}$ can be thus written as:(40)$$C_{X_{i}^{j}}\left(X_{h}^{k}\right)=P\left(X_{h}^{k}|X_{i}^{j}\right)=\frac{P(X_{h}^{k},X_{i}^{j})}{P(X_{i}^{j})}$$

We can then write down expressions for Markov conditional independence as follows:(41)$$X_{i}^{j}⊥\;\;\;⊥X_{h}^{k}|X_{l}^{m}\Rightarrow P\left(X_{i}^{j},X_{h}^{k}|X_{l}^{m}\right)=P\left(X_{i}^{j}|X_{l}^{m}\right)\times P\left(X_{h}^{k}|X_{l}^{m}\right)$$

Following Bayes’ theorem, CI conditions –in this case– will be of the form:(42)$$P\left(X_{i}^{j},X_{h}^{k}|X_{l}^{m}\right)=\frac{P(X_{i}^{j},X_{l}^{m})}{P(X_{l}^{m})}\times \frac{P(X_{h}^{k},X_{l}^{m})}{P(X_{l}^{m})}=\frac{P\left(X_{i}^{j},X_{l}^{m}\right)\times P\left(X_{h}^{k},X_{l}^{m}\right)}{{P(X_{l}^{m})}^{2}}$$

Equation (Equation 42) is useful since in large scale data applications is computationally cheaper to work with joint and marginal probabilities rather than conditionals.

Now let us consider the case of conditional independence given several conditional variables. The case for CI given two variables could be written—using conditional kernels—as follows:(43)$$X_{i}^{j}⊥\;\;\;⊥X_{h}^{k}|X_{l}^{m},X_{n}^{o}\Rightarrow P\left(X_{i}^{j},X_{h}^{k}|X_{l}^{m},X_{n}^{o}\right)=P\left(X_{i}^{j}|X_{l}^{m},X_{n}^{o}\right)\times P\left(X_{h}^{k}|X_{l}^{m},X_{n}^{o}\right)$$

Hence,
(44)$$P\left(X_{i}^{j},X_{h}^{k}|X_{l}^{m},X_{n}^{o}\right)=C_{X_{l}^{m},X_{n}^{o}}\left(X_{i}^{j}\right)\times C_{X_{l}^{m},X_{n}^{o}}\left(X_{h}^{k}\right)$$

Using Bayes’ theorem,
(45)$$P\left(X_{i}^{j},X_{h}^{k}|X_{l}^{m},X_{n}^{o}\right)=\frac{P(X_{i}^{j},X_{l}^{m},X_{n}^{o})}{P(X_{l}^{m},X_{n}^{o})}\times \frac{P(X_{h}^{k},X_{l}^{m},X_{n}^{o})}{P(X_{l}^{m},X_{n}^{o})}$$
(46)$$P\left(X_{i}^{j},X_{h}^{k}|X_{l}^{m},X_{n}^{o}\right)=\frac{P\left(X_{i}^{j},X_{l}^{m},X_{n}^{o}\right)\times P\left(X_{h}^{k},X_{l}^{m},X_{n}^{o}\right)}{{P(X_{l}^{m},X_{n}^{o})}^{2}}$$

In order to generalize the previous results to CI relations given an arbitrary set of conditionals, let us consider the following *sigma-algebraic* approach:

Let $\Sigma_{ih}^{jk}$ be the σ-algebra of all subsets of *X* that do not contain $X_{i}^{j}$ or $X_{h}^{k}$. If we consider the contravariant index i∈α with i=1,2,…,N and the covariant index j∈β with j=1,2,…,L, then there are $M=\frac{NL}{2}$ such σ-algebras in *X* (let us recall that TMFs are *undirected* graphical models).

A relevant problem for network reconstruction is that of establishing the more general Markov pairwise CI conditions, i.e., the CI relations for every edge not drawn in the graph. Two arbitrary nodes $X_{i}^{j}$ and $X_{h}^{k}$ are conditionally independent given the rest of the graph iff:(47)$$X_{i}^{j}⊥\;\;\;⊥X_{h}^{k}\;|\;\Sigma_{ih}^{jk}\Rightarrow P\left(X_{i}^{j},X_{h}^{k}\;|\;\Sigma_{ih}^{jk}\right)=P\left(X_{i}^{j}\;|\;\Sigma_{ih}^{jk}\right)\times P\left(X_{h}^{k}\;|\;\Sigma_{ih}^{jk}\right)$$

By using conditional kernels, the recursive relations and Bayes’ theorem it is possible to write down M expressions of the form:(48)$$P\left(X_{i}^{j},X_{h}^{k}\;|\;\Sigma_{ih}^{jk}\right)=\frac{P\left(X_{i}^{j},\Sigma_{ih}^{jk}\right)\times P\left(X_{h}^{k},\Sigma_{ih}^{jk}\right)}{{P(\Sigma_{ih}^{jk})}^{2}}$$

The family of Equations (Equation 48) represent the CI relations for all the non-existing edges in the hypergraph $\hat{G}$, i.e., every pair of nodes $X_{i}^{j}$ and $X_{h}^{k}$ not-connected in $\hat{G}$ must be conditionally independent given the rest of the nodes in the graph. These expression may serve to implement exact tests or optimization strategies for *graph reconstruction* and/or *graph sparsification* in applications considering a mutual information threshold $I_{0}$ as in Equation (Equation 32).

In brief, for every node pair with a mutual information value lesser than $I_{0}$, the presented graph reconstruction approach will not draw an edge, hence implying CI between the two nodes given the rest. Such CI condition may be tested on the data to see whether it holds or the threshold itself can be determined by resorting to optimization schemes (e.g., error bounds) in Equation (Equation 48).

## 4. Graph Theoretical Features and Multilinear Structure

Once the probabilistic properties of TMFs have been set, it may be fit to briefly present some of their graph theoretical features, as well as some preliminaries as to the reasons to embed hyperadjecency matrices into multilayer adjacency tensors. Given that TMFs are indeed PGMs, some of their graph characteristics will result relevant here.

Since the work by De Domenico and coworkers [5] covers in great detail how the multilinear structure of the multilayer adjacency tensor allows the calculation of these quantities—usually as projection operations—we will only mention connectivity degree vectors since these are related with the size of the TMF dependency neighborhoods.

Let us recall multilayer adjacency tensors, as defined in Equations (Equation 34) and (Equation 35). To ease presentation, we will work with the unweighted tensor $A_{\beta \delta }^{\alpha \gamma }$ (Equation (Equation 34)). The *multidegree centrality vector*
$K^{\alpha }$ which contains the connectivity degrees of the nodes spanning different layers can be written as follows:(49)$$K^{\alpha }=A_{\beta \delta }^{\alpha \gamma }\;U_{\gamma }^{\delta }\;u^{\beta }$$

Here $U_{\gamma }^{\delta }$ is a rank 2 tensor that contains a 1 in every component and $u^{\beta }$ is a rank 1 tensor that contains a 1 in every component—these quantities are called 1—tensors by De Domenico and coworkers [5]. It can be shown that $K^{\alpha }$ is indeed given by the sums of the *connectivity degree vectors*
$k^{\alpha }$ corresponding to all different layers:(50)$$K^{\alpha }=\sum_{h=1}^{L}\sum_{k=1}^{L}\;k^{\alpha }\left(hk\right)$$

$k^{\alpha }\left(hk\right)$ is the vector of connections that nodes in the set α=1,2,…,N in layer *h* have to any other nodes in layer *k*. Whereas $K^{\alpha }$ is the vector with connections in all the layers. Appropriate projections will yield measures such as the size of the neighborhood to a given vertex $|N_{X_{i}^{j}}|$, the size of its Markov blanquet $\left|mb(X_{i}^{h})\right|$, or other similar quantities.

## 5. Specific Applications

After having considered some of the properties of this class of Tensor Markov Fields, it may become evident that aside from purely theoretical importance, there is a number of important applications that may arise as probabilistic graphical models in tensor valued problems, among the ones that are somewhat evident are the following:The analysis of multidimensional biomolecular networks such as the ones arising from multi-omic experiments (For a real-life example, see Figure 4) [8,9,10];Probabilistic graphical models in computer vision (especially 3D reconstructions and 4D [3D+time] rendering) [11];The study of fracture mechanics in continuous deformable media [12];Probabilistic network models for seismic dynamics [13];Boolean networks in control theory [14].

Some of these problems are being treated indeed as multiple instances of Markov fields or as multipartite graphs or hypergraphs. However, it may become evident that when random variables *across layers* are interdependent (which is often the case), the definitions of potentials, cliques and partition functions, as well as the conditional statistical independence features become manageable (and in some cases even meaninful) under the presented formalism of Tensor Markov Fields.

## 6. Conclusions

Here we have presented the definitions and fundamental properties of Tensor Markov Fields, i.e., random Markov fields over tensor spaces. We have proved –by extending the results of Dobruschin, Hammersley and Clifford to such tensor valued fields– that tensor Markov fields are indeed Gibbs fields whenever strictly positive probability measures are considered. We also introduced a class of tensor Markov fields obtained by using information theoretical statistical dependence measures inducing local and global Markov properties, and show how these can be used as probabilistic graphical models in multi-context environments much in the spirit of the so-called multilayer network approach. Finally, we discuss the convenience of embedding tensor Markov fields in the multilinear tensor representation of multilayer networks. 

## Figures and Tables

**Figure 1 entropy-22-00451-f001:**
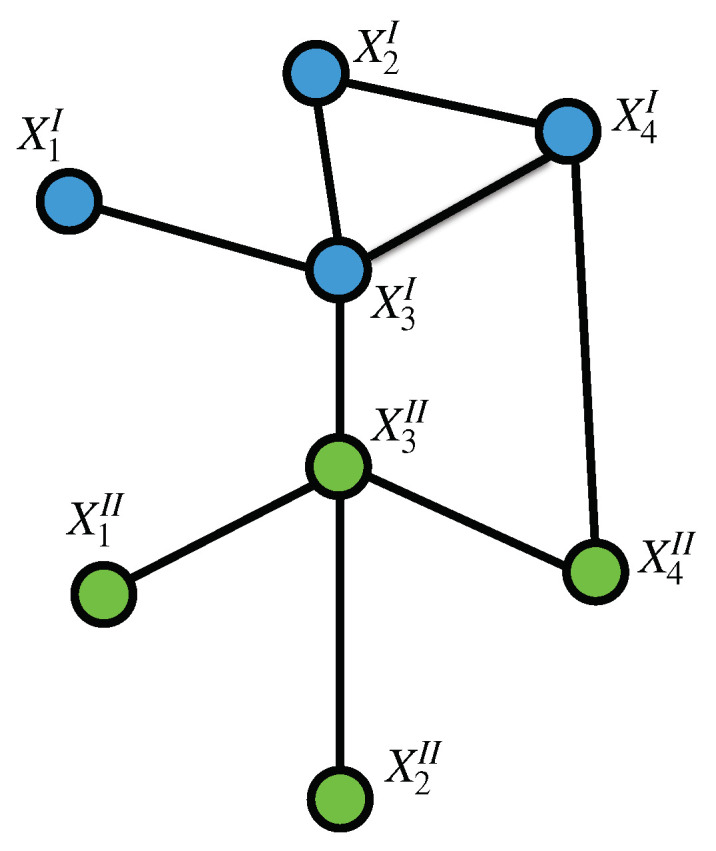
A Tensor Markov Field: represented as a multilayer graph spanning over $X_{i}^{j}$ with i={1,2,3,4} and j={I,II}. To illustrate, layer *I* is colored in blue and layer II is colored green.

**Figure 2 entropy-22-00451-f002:**
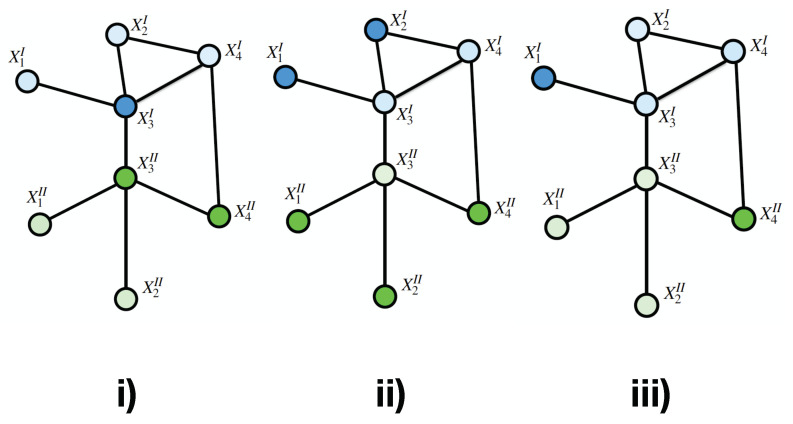
Three different configurations of a Tensor Markov Fieldpanels (**i**), (**ii**) and (**iii**) present different configurations or *states* of the TMF. Labels are represented by color intensity.

**Figure 3 entropy-22-00451-f003:**
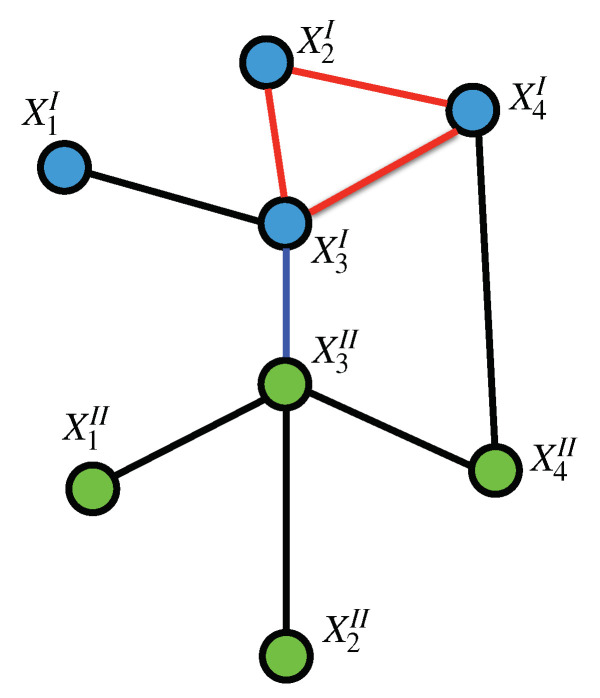
Cliques on a Tensor Markov Field: The set $\left\{X_{2}^{I},X_{3}^{I},X_{4}^{I}\right\}$ forms an intra-layer 2-clique (as marked by the red edges, all on layer *I*), the set $\left\{X_{3}^{I},X_{3}^{II}\right\}$ forms an inter-layer 1-clique (marked by the blue edge connecting layers *I* and II). However, the set $\left\{X_{3}^{I},X_{3}^{II},X_{4}^{I},X_{4}^{II},\right\}$ is not a clique since there are no edges between $X_{3}^{I}$ and $X_{4}^{II}$ nor between $X_{3}^{I}$ and $X_{4}^{II}$.

**Figure 4 entropy-22-00451-f004:**
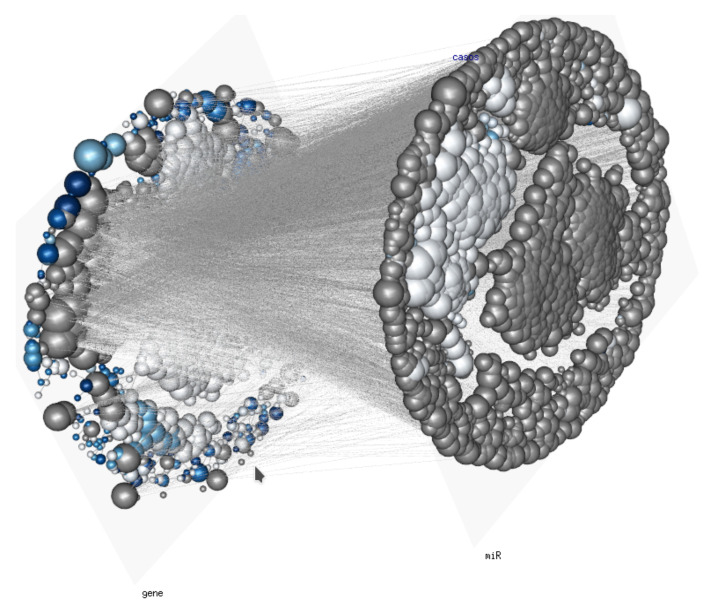
Gene and microRNA regulatory network: A Tensor Markov Field depicting the statistical dependence of genome wide gene and microRNA (miR) on a human phenotype. Edge width is given by the mutual information $I^{†}\left(X_{i}^{j},X_{h}^{k}\right)$ between expression levels of genes (layer *j*) and miRs (layer *k*) in a very large corpus of RNASeq samples, vertex size is proportional to the *degree*, i.e., the size of the node’s neighborhood, $N_{X_{i}^{j}}$.

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
