# Peer review of "On a Class of Tensor Markov Fields"

_entropy, 2020, doi:10.3390/e22040451_

Round 1

Reviewer 1 Report

The present work generalizes, to tensor-valued fields, previous well-known results [1]. A convenient potential function is defined on the tensor fields distribution, yielding a Gibbs potential form. The authors discuss the informational properties of the system and the conditional independence for tensor Markov fields (TMF). Interesting results are obtained, such as eq 48, which presents the conditional independence condition for the non-existing edges in the hypergraph where the TMF is defined.

The paper is well written and clear. There are minor mistakes that the author might want to correct such as “pont” in line 30. Also, energy (l51) and potential energy (l60) are better defined as dimensionless energy & dimensionless potential energy, respectively.

The calculations are hard to follow but, as far as I could check, they seem correct. However, the manuscript would benefit greatly by including figures, in special in the general definitions section. An illustration of the TMF multilayered graph would help the reader to visualize the problem.

In my opinion, the manuscript can be accepted for publication on Entropy, after the authors include the corrections and suggestions above.

Reviewer 2 Report

The author discusses Tensor Markov Fields, i.e random Markov fields over tensor spaces, extending previous results by Dobruschin, Hammersley and Clifford. He shows that tensor Markov fields are Gibbs fields whenever strictly positive probability measures are considered. Although this is not particularly surprising, it seems both interesting and correct. The author suggests that a class of these tensor Markov fields can be used as probabilistic graphical models in multi-context environments much in the spirit of the so-called multilayer network approach. It would be nice to see in the future some examples of specific applications of this.
